# The *Neisseria gonorrhoeae* type IV pilus promotes resistance to hydrogen peroxide- and LL-37-mediated killing by modulating the availability of intracellular, labile iron

**Linda I. Hu**⊙, **Elizabeth A. Stohl**⊙, **H Steven Seifert**⊙*

Department of Microbiology-Immunology, Northwestern University Feinberg School of Medicine, Chicago, Illinois, United States of America

⊙ These authors contributed equally to this work.
* h-seifert@northwestern.edu

**Data Availability Statement:** All relevant data are within the manuscript and its Supporting Information files.

## Abstract

The *Neisseria gonorrhoeae* Type IV pilus is a multifunctional, dynamic fiber involved in host cell attachment, DNA transformation, and twitching motility. We previously reported that the *N. gonorrhoeae* pilus is also required for resistance against hydrogen peroxide-, antimicrobial peptide LL-37-, and non-oxidative, neutrophil-mediated killing. We tested whether the hydrogen peroxide, LL-37, and neutrophil hypersensitivity phenotypes in non-piliated *N. gonorrhoeae* could be due to elevated iron levels. Iron chelation in the growth medium rescued a nonpiliated *pilE* mutant from both hydrogen peroxide- and antimicrobial peptide LL-37-mediated killing, suggesting these phenotypes are related to iron availability. We used the antibiotic streptonigrin, which depends on free cytoplasmic iron and oxidation to kill bacteria, to determine whether piliation affected intracellular iron levels. Several non-piliated, loss-of-function mutants were more sensitive to streptonigrin killing than the piliated parental strain. Consistent with the idea that higher available iron levels in the under- and non-piliated strains were responsible for the higher streptonigrin sensitivity, iron limitation by desferal chelation restored resistance to streptonigrin in these strains and the addition of iron restored the sensitivity to streptonigrin killing. The antioxidants tiron and dimethylthiourea rescued the *pilE* mutant from streptonigrin-mediated killing, suggesting that the elevated labile iron pool in non-piliated bacteria leads to streptonigrin-dependent reactive oxygen species production. These antioxidants did not affect LL-37-mediated killing. We confirmed that the *pilE* mutant is not more sensitive to other antibiotics showing that the streptonigrin phenotypes are not due to general bacterial envelope disruption. The total iron content of the cell was unaltered by piliation when measured using ICP-MS suggesting that only the labile iron pool is affected by piliation. These results support the hypothesis that piliation state affects *N. gonorrhoeae* iron homeostasis and influences sensitivity to various host-derived antimicrobial agents.

**Funding:** This work was supported by National Institute of Allergy and Infectious Diseases (US) grants R01 AI146073 and R37 AI033493 to HSS. Metal analysis was performed at the Northwestern University Quantitative Bio-element Imaging Center generously supported by NASA Ames Research Center Grant NNA04CC36G. The funders had no role in study design, data collection and analysis, decision to publish, or preparation of the manuscript.

**Competing interests:** The authors have declared that no competing interests exist.

## Author summary

*Neisseria gonorrhoeae* is a bacterium that causes the sexually transmitted infection, gonorrhea. The bacteria express a fiber on their surface called a pilus that mediates many interactions of the bacterial cell with host cells and tissues. The ability to resist killing by white cells is one important ability that *N. gonorrhoeae* uses to allow infection of otherwise healthy people. We show here that the pilus help resist white cell killing by modulating the levels of iron within the bacterial cell.

## Introduction

*Neisseria gonorrhoeae* is an obligate human pathogen that is the sole cause of the sexually transmitted infection, gonorrhea. Over 87 million gonorrhea cases are reported globally each year [1] and 616,000 cases were reported to the Centers for Disease Control and Prevention in the United States in 2019, the highest number since 1991 [2]. Combined with the fact that there are strains resistant to commonly administered antibiotics, gonorrhea has become an increasingly alarming concern to public health [3–5], and has led to the CDC labeling *N. gonorrhoeae* as an urgent public health threat.

Colonization primarily occurs in the genitourinary tract but can also infect the ocular, nasopharyngeal, and rectal mucosa [6–8]. Purulent exudate, a genital secretion of a mix of fluid, bacteria, and neutrophils (or polymorphonuclear leukocytes, PMNs), and dysuria, pain during urination, are classic indications of symptomatic gonococcal infection [9]. Colonization with no observable symptoms can also occur in both men and women [10]. In women, an untreated infection can spread from the cervix to other areas of the reproductive organ to cause pelvic inflammatory disease. In men, infections can cause epididymitis which is inflammation of the epididymis. These mucosal gonococcal infections can also enter the bloodstream to develop disseminated gonococcal infection and can lead to dermatitis, arthritis, endocarditis, and neurological issues [11]. Additionally, one's risk of HIV acquisition and transmission increases if the person is co-infected with *N. gonorrhoeae* [12–14].

The ability of *N. gonorrhoeae* to resist host immune responses is critical for pathogenesis. *N. gonorrhoeae* are equipped with multiple antioxidant mechanisms whose products protect *N. gonorrhoeae* against oxidative damage and neutrophils [15], including *ngo1686*, renamed to *mpg* for M23B metalloprotease active against peptidoglycan, *recN* [16,17], and a transcriptional response that consists of hundreds of genes when exposed to hydrogen peroxide [17,18]. Additionally, disrupting *N. gonorrhoeae* genes whose products have antioxidant functions (*katA*, *sodB*, *ccp*, or *mntABC*), as single mutants or in combination, did not affect neutrophil-mediated killing [19,20] or infection in a vaginal mouse model [21]. However, anti-gonococcal neutrophil activities are mainly non-oxidative; therefore, these gene products are likely acting in response to other sources of reactive oxygen species (ROS). In the absence of an oxidative burst, neutrophils from anoxic conditions [22] or from patients with chronic granulomatous disease [19,23], an immunodeficiency that renders neutrophils unable to elicit an oxidative burst due to mutations in NADPH oxidase, are still cytotoxic to *N. gonorrhoeae*. Even in the presence of a bactericidal oxidative response, live *N. gonorrhoeae* can suppress neutrophil ROS production [24]. These results support the conclusion that PMN anti-gonococcal activities are mainly non-oxidative and it has been proposed that *N. gonorrhoeae* genes that are involved in antioxidant responses are protective in certain oxidative conditions, such as from the mucosal epithelium, vaginal lactobacilli, and cellular respiration [25].

In addition to non-oxidative antibacterial effectors like cathepsin proteinases and peptido-glycan-targeting lysozyme, neutrophils release short, positively charged antimicrobial peptides (AMPs). LL-37 is an alpha-helical cathelicidin produced from PMNs that can kill *N. gonor-rhoeae*, and the MtrCDE efflux pump system decreases AMP susceptibility [26]. The mechanism of how AMPs kill cells is not known but may depend on an interaction of the peptide with the bacterial membrane. The cationic nature of the peptide can mediate an electrostatic interaction with the negatively charged bacterial outer membrane and, subsequently, the cytoplasmic membrane. Once a critical local concentration of the peptide accumulates on the membrane, these amphipathic peptides can disrupt the cell membrane through the formation of holes that lead to cell death [27]. However, there is also evidence that AMPs have intracellular targets that are important for the AMP mechanism of action [27]. The increased sensitivity to LL-37 killing in efflux pump mutants raises the possibility that LL-37 has activity within the bacterial cytoplasm [28]. In support of this hypothesis is the report that LL-37 can gain access to the bacterial cytoplasm where it can bind to DNA, interrupt DNA replication, and cause DNA mutations [29]. LL-37 can also bind DNA and increase the viscosity of the cytoplasm [30]. Therefore, LL-37 may have different modes of action against *N. gonorrhoeae*.

Type IV pili are critical in establishing infection by mediating adherence of the bacterial cells to the host mucosal epithelium. Type IV pili are dynamic appendages that can extend and retract, are critical in microcolony formation, host adherence, twitching motility, and natural DNA competence [31]. The pilus undergoes phase variation and antigenic variation, processes that contribute to adaptive immune evasion and can change the expression level of the pilus.

Pili are also important for gonococcal resistance to hydrogen peroxide-, LL-37-, and neutrophil-mediated killing [32]. In characterizing the transcriptional response to hydrogen peroxide, we reported that the gene in the *ngo1686* locus *mpg* was upregulated by oxidative damage [17]. In multiple *mpg* loss-of-function mutants, *N. gonorrhoeae* cells were underpiliated and sensitive to hydrogen peroxide, suggesting that the pilus promoted hydrogen peroxide resistance [33]. In addition, a predominantly Opa-negative, *pilE* mutant was more sensitive to hydrogen peroxide-, LL-37-, and nonoxidative neutrophil-mediated killing than the isogenic piliated strain [32]. A double *mpg/pilE* mutant was equally sensitive as the *pilE* mutant, supporting that these genes are part of the same pathway. Here, we follow up on those phenotypes to report that the pilus-mediated resistance to hydrogen peroxide and LL-37 killing is due to a reduced level of intracellular available iron pool, suggesting that pilus expression modulates iron homeostasis within the bacterial cell to protect against the antibacterial action of neutrophils.

## Results

### Iron chelation rescues the *pilE* mutant from hydrogen peroxide- and LL-37-mediated killing

We previously reported that a *pilE* mutant is more sensitive to various antimicrobial agents than the FA1090 piliated strain [32]. We confirmed that the *pilE* mutant was more sensitive to both hydrogen peroxide and LL-37 than the parental strain (**Fig 1A** and **1B**). We reasoned that there could be increased sensitivity to hydrogen peroxide through iron-generating hydroxyl radicals using Fenton chemistry. To determine whether the available iron affects sensitivity to hydrogen peroxide- and LL-37-mediated killing in the *pilE* mutant, we compared the effect of iron chelation on survival using desferal (also named desferrioxamine) [34]. When we added desferal to the growth medium, relative survival of the *pilE* mutant against hydrogen peroxide increased about five-fold when comparing an average relative survival of 0.79 with 10 mM desferal to 0.15 without desferal (**Fig 1C**). Desferal also affected the sensitivity of the *pilE* mutant to LL-37, increasing resistance by seven-fold when comparing an average relative

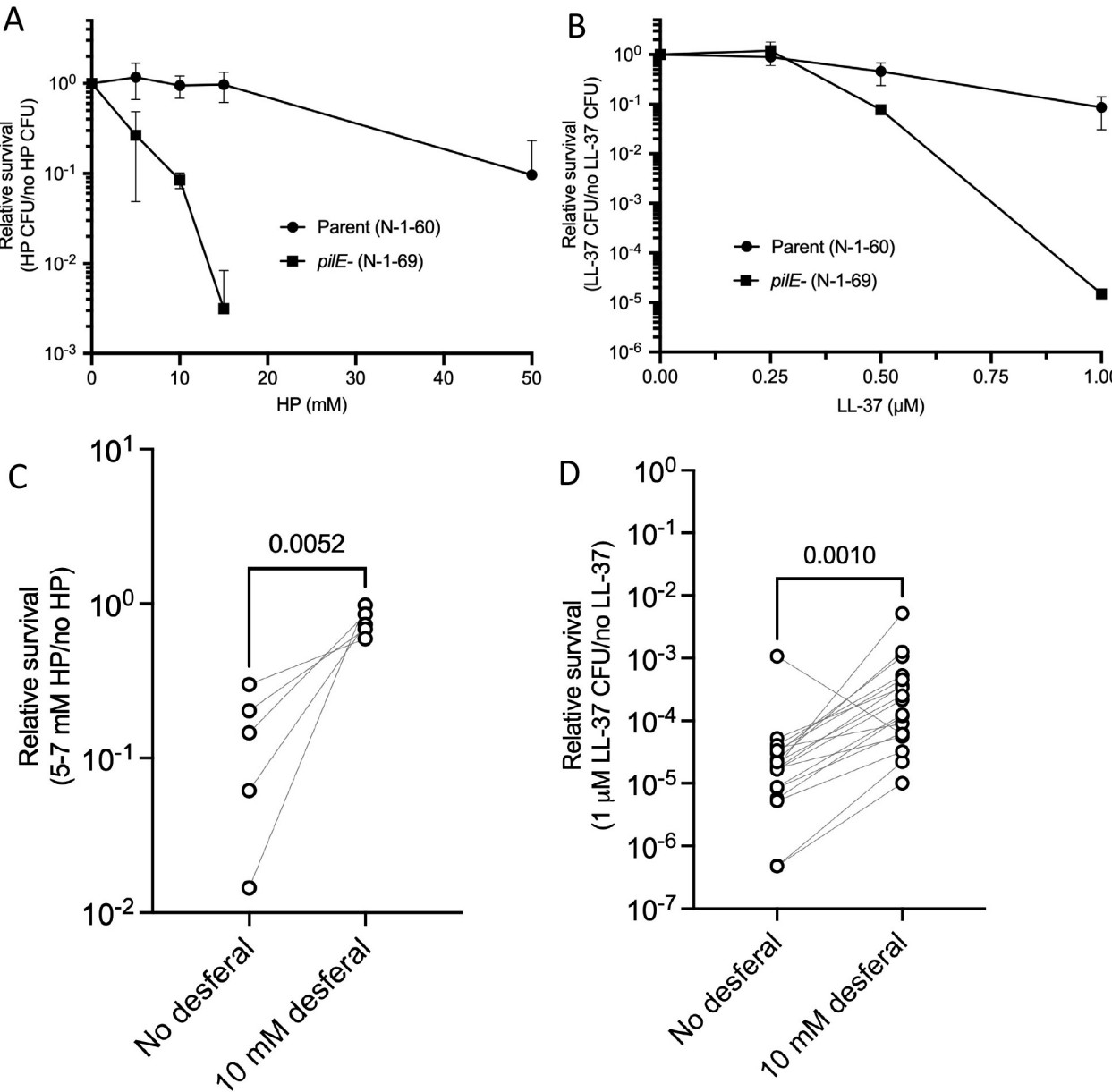

**Fig 1. Iron chelation rescues the *pilE* mutant from HP- and LL-37-mediated killing.** Relative survival of the parental strain (N-1-60) and the isogenic *pilE* mutant (N-1-69) after (A) hydrogen peroxide (HP) treatment or (B) LL-37 exposure. C. Relative survival of the FA1090 *pilE* mutant after pre-treatment with desferal before HP exposure. D. LL-37 sensitivities of the *pilE* mutant after desferal treatment. The data are presented with lines to indicate the matched pairs and analyzed by a paired t test in panel C and a Wilcoxon matched pairs signed-rank test in panel D with GraphPad Prism.

survival of $5.7 \times 10^{-4}$ with 10 mM desferal to $7.8 \times 10^{-5}$ without desferal (**Fig 1D**). We tested the effect of desferal on the parental piliated strain and have found that there was no difference in sensitivity to either hydrogen peroxide or LL-37 (**S1 Fig**).

## Non-piliated cells are hypersensitive to streptonigrin in an iron-dependent manner

Sensitivity of the bacterial cells to the antibiotic streptonigrin is influenced by differences in bacterial iron levels. This antibiotic presently has limited clinical use due to its toxicity and

side-effects [35]. Streptonigrin is a hydrophobic compound that is poorly soluble in water and often dissolved in an organic solvent. Streptonigrin can, therefore, diffuse across the membrane, bind to ferrous iron in the bacterial cytoplasm, and generate cytotoxic reactive oxygen species [36]. As a result, the amount of streptonigrin required to kill a bacterial cell is proportional to the intracellular, labile iron pool [37–39]. We tested whether a *pilE* mutant was more sensitive to streptonigrin compared to the piliated parental strain (**Fig 2A**). The nonpiliated mutant was more sensitive to streptonigrin killing at all levels of antibiotic tested and increasing the concentration of streptonigrin resulted in a greater differential killing between the strains (**Fig 2A**). For example, the *pilE* mutant showed over 55 thousand-fold greater sensitivity to 1.6 μM streptonigrin than the piliated parent.

Because the loss of membrane integrity in *Vibrio cholerae* increases oxidative stress and labile, intracellular iron [40], we tested whether the differential sensitivity of the nonpiliated cells could be due to a general leakiness of the bacterial envelope. We assayed the sensitivity of the *pilE* mutant to several antibiotics representing different classes. While the *pilE* mutant displayed some differences in sensitivities to some antimicrobials relative to the piliated parent strain, the absence of piliation did not lead to a hypersensitivity to other antibiotics (**Table 1**), similar to the results from a previous study [41].

To test whether streptonigrin hypersensitivity is due to iron, we treated the *pilE* mutant or the parent strain with either streptonigrin alone, desferal alone, or desferal before streptonigrin treatment, and determined the relative survival. Iron-chelation by desferal had a considerable effect on streptonigrin-mediated killing of the *pilE* mutant (**Fig 2B**) and the parental strain (**Fig 2C**). These results are consistent with a previous report showing desferal reduced killing by streptonigrin [42].

Desferal has a high affinity for iron (Km ~$10^{-26}$) but can also complex with other metal ions with lower affinities [34]. To determine whether desferal chelating other metals in the media could be responsible for desferal-mediated rescue in the *pilE* mutant, we tested the effect of adding various metals, including iron, on desferal-mediated rescue from streptonigrin killing. We preincubated desferal with increasing concentrations of iron(III) chloride, manganese(II) chloride, zinc(II) chloride, and magnesium(II) chloride and exposed cells to these desferal-metal mixtures before streptonigrin killing. While we did observe that excess iron promoted streptonigrin resistance, iron also interfered with desferal-dependent rescue from streptonigrin killing in the *pilE* mutant (**Figs 2D** and **S2A**). We detected similar effects of excess magnesium (**S2B Fig**); however, the fold reduction in desferal rescue was greater with iron (over 100-fold) compared to magnesium (less than 10-fold). While zinc and manganese increased resistance to streptonigrin, they did not affect desferal-dependent streptonigrin rescue (**S2C** and **S2D Fig**).

## Pilus expression promotes streptonigrin resistance

While the piliated and nonpiliated strains we used are isogenic since the nonpiliated strain was derived from the piliated parent, it remained a possibility that second site mutation(s) could be responsible for the differential sensitivity phenotypes [43]. To directly test whether *pilE* expression alone affected the level of streptonigrin sensitivity, we used an IPTG-inducible *pilE* construct where the *lacIOP* regulatory region was inserted in the 5' untranslated region of *pilE*, between the *pilE* promoter and the open reading frame [44] (**Fig 2E**). In the absence of *pilE* induction, cells were hypersensitive to streptonigrin-mediated killing like the *pilE* mutant. When piliation was restored by adding IPTG into the growth medium, killing was reduced by over 1000-fold. We confirmed that as previously reported, 1 mM IPTG increased transformation competence by over seven hundred-fold confirming the differential pilus expression (**S3 Fig**). These results confirm that it is pilin expression that affects streptonigrin resistance and this effect was most likely due to type IV pilus expression.

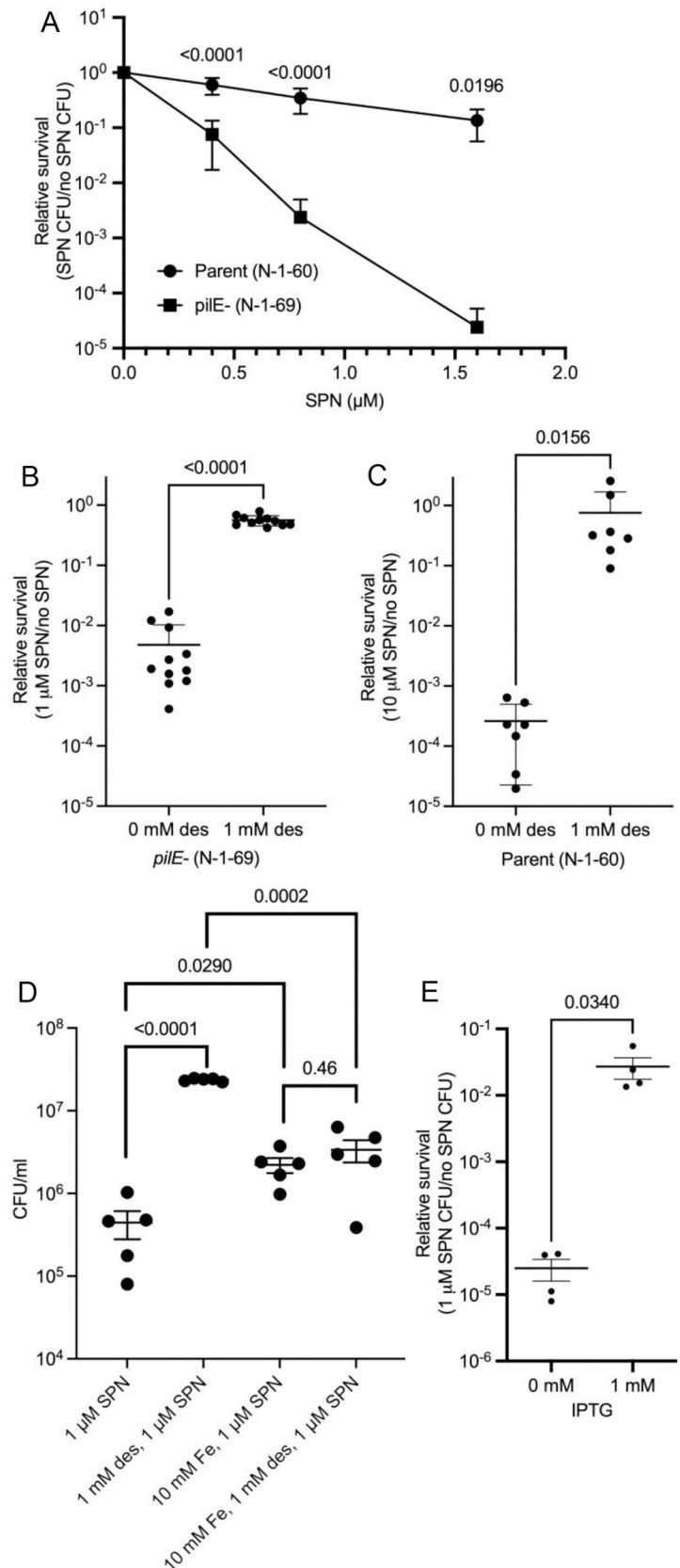

**Fig 2. Pilus expression mediates resistance to streptonigrin killing in an iron-dependent manner.** A. The effect of streptonigrin (SPN) on the parental strain FA1090 and the *pilE* mutant. Strains were treated with DMSO or streptonigrin and the relative survival was determined (n = 15). The average and standard error are shown. The data were analyzed by ANOVA and p values are indicated. B. The effect of desferal (des) on streptonigrin resistance in the *pilE* mutant (n = 11). C. The effect of des on streptonigrin resistance in the parent (n = 7). D. The effect of ferric chloride on desferal-mediated rescue from streptonigrin killing in the *pilE* mutant. E. The effect of *pilE* expression on streptonigrin resistance. IPTG in the growth medium was used to induce chromosomal *pilE* expression prior to treatment with streptonigrin (n = 4). Relative survival was determined by plating CFUs from the various conditions. Averages and standard error of independent biological replicates are shown and significance measured using a paired t-test in panel B, a Wilcoxan matched pairs signed-rank test in panel C, a Tukey's multiple comparison test in panel D, and a paired t test in panel E with GraphPad Prism.

## PilE-dependent resistance to streptonigrin occurs in multiple *N. gonorrhoeae* strains

To determine whether this pilus-dependent phenotype was specific to the FA1090 strain, we tested if two other common lab strains of *N. gonorrhoeae* exhibit hypersensitivity to streptonigrin in nonpiliated cells. The *pilE* gene was deleted in isolates F62 and FA19, and each parental strain and isogenic *pilE* mutant was treated with increasing concentrations of streptonigrin. Both the F62 and FA19 strains are hypersensitive to streptonigrin in the absence of *pilE* and FA19 was more sensitive than F62 (**Fig 3**).

## Streptonigrin kills the *pilE* mutant in a ROS-dependent manner and LL-37 kills in a ROS-independent manner

Elevated labile iron pools can lead to greater ROS through the Fenton chemistry. To determine whether intracellular levels of ROS are elevated in the *pilE* mutant, we tested several ROS scavengers that quench various ROS species. Consistent with a previous report that anaerobically grown *N. gonorrhoeae* cells are resistant to streptonigrin [42], we found that the ROS scavengers dimethylthiourea (DMTU) and tiron can rescue the *pilE* mutant from streptonigrin-dependent killing. There was an approximately 330-fold increase in relative survival to streptonigrin when cultures were treated with 15 mM DMTU and a 150-fold increase when treated with 0.5 mM tiron (**Fig 4**). We tested whether DMTU or tiron affected LL-37-mediated killing in the *pilE* mutant but did not observe any rescue (**S4 Fig**). These results suggest that the level of iron in non-piliated cells affect sensitivity to streptonigrin through a ROS-dependent mechanism, while the iron-dependent LL-37 killing occurs through a ROS-independent manner.

## Total iron content is unaffected by piliation

Due to the iron-dependent, differential sensitivities of the *pilE* mutant to streptonigrin, hydrogen peroxide, and LL-37, we tested the hypothesis that piliation affected total iron levels in both the parental strain and *pilE* mutant by comparing the level of iron measured by ICP-MS [45,46]. The parental strain and the *pilE* mutant showed no significant difference in total iron

**Table 1. Antibiotic sensitivity of the piliated parent and a *pilE* mutant.**

| Antibiotic | Ampicillin | Polymyxin B | Tetracycline | Naladixic acid | Rifampicin |
|---|---|---|---|---|---|
| Target | Cell wall | Cell wall | Translation | DNA replication | Transcription |
| Parent | 0.13 | 48–64 | 0.13 | 0.50 | 0.02 |
| Δ*pilE* | 0.125–0.19 | 64.00 | 0.125–0.19 | 0.50 | 0.02 |

The minimum inhibitory concentrations ranges from various antimicrobials on three biological replicates of the parental strain (N-1-60) and the *pilE* mutant (N-1-69) using Etest strips (µg/ml).

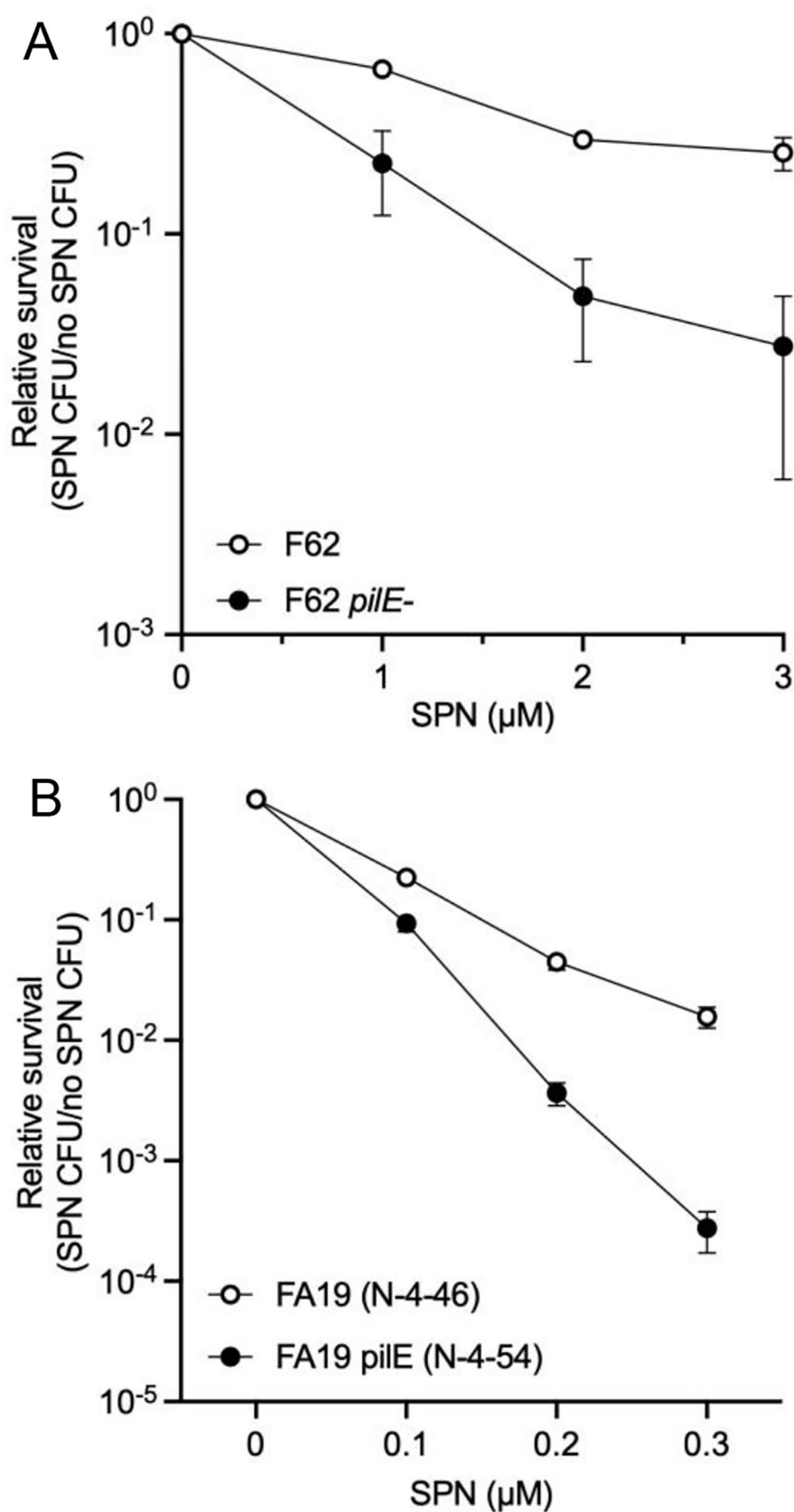

**Fig 3. The *N. gonorrhoeae* strain background impacts the PilE-dependent streptonigrin resistance phenotype.** A. The F62 (n = 3) and B. FA19 (n = 4) strains and their isogenic *pilE* mutants were treated with increasing concentrations of streptonigrin. The average and standard error of biological replicates are shown.

normalized to total protein (parent = 0.127 average Fe/total protein, SD = 0.02; *pilE* mutant = 0.126 average Fe/total protein, SD = 0.014; *pilE* mutant/parent ratio = 0.995, p-value = 0.96 Student t-test). These results indicated that *pilE* hypersensitivity to streptonigrin, hydrogen peroxide, and LL-37 is not due to differential levels of total intracellular iron.

## Discussion

Iron is a critical metal in the competition between bacteria and their hosts [47]. While iron is a necessary co-factor of many enzymes, it can exacerbate the toxic effects of oxygen through the Fenton reaction. We had previously reported that piliation provides *N. gonorrhoeae* resistance to hydrogen peroxide, LL-37, and neutrophils [32]. We now have shown that both hydrogen peroxide and LL-37 sensitivity are dependent on the cells being iron-replete and that hydrogen peroxide sensitivity is due to the development of ROS. LL-37 sensitivity is dependent on iron; however, iron is mediating LL-37 non-oxidative killing of non-piliated cells through unknown factors.

We propose that the differential sensitivity to streptonigrin, hydrogen peroxide, and LL-37 is from the effect of pilus expression on modulating the intracellular labile iron pool. This

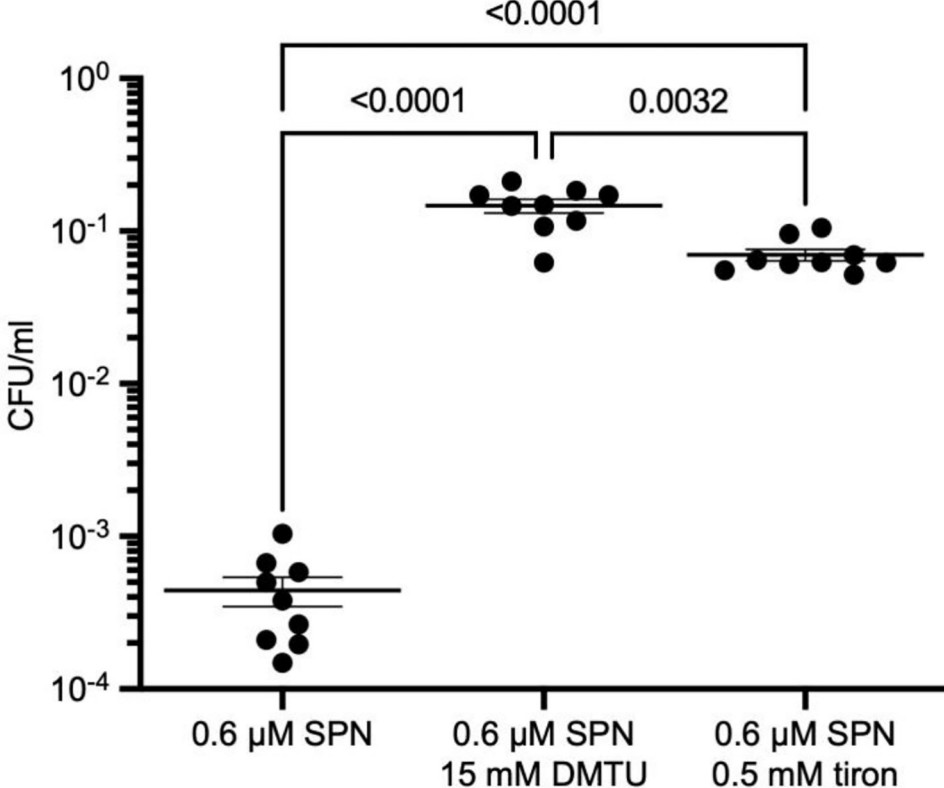

**Fig 4. ROS scavengers DMTU and tiron protect against SPN-mediated killing in *pilE* mutant.** Streptonigrin sensitivity of the *pilE* mutant after pre-treatment with either DMTU or tiron. The mean and standard error of the mean are plotted and analyzed with one-way ANOVA.

elevated level of available iron in non-piliated cells would favor the hydroxyl-radical-forming Fenton reaction and alter the sensitivity threshold to oxidative damage. This hypothesis is supported by the fact that *pilE* mutant survival from streptonigrin killing is reduced by the ROS scavengers DMTU and tiron. DMTU is regarded as a hydrogen peroxide and hydroxyl radical scavenger [48,49], while tiron is characterized as a superoxide anion scavenger [50], which are all molecules that are produced after hydrogen peroxide treatment. In a study comparing transcriptional changes in piliated and nonpiliated *N. gonorrhoeae*, *recN* and *ngo1769* (a cytochrome *c* peroxidase) showed approximately two times higher expression in piliated gonococci [51], suggesting that increased expression of proteins with antioxidant activities may be involved in pilin-dependent resistance to oxidative killing by hydrogen peroxide and streptonigrin.

Interestingly, while LL-37 also exhibited enhanced, iron-dependent killing of nonpiliated strains, the bactericidal activity was unaffected by DMTU or tiron treatment, suggesting that its antimicrobial activity is non-oxidative. A similar iron-dependent toxic effect of LL-37 was found for *Pseudomonas aeruginosa* [52]. The authors suggested that LL-37 promoted the influx of iron into the cells and that promoted DNA damage through Fenton reactions. This would be a possible explanation if piliation modulates iron import and, in the absence of the pilus, iron influx is uninhibited, leading to differential killing in the *pilE* mutant. However, if piliation affected membrane permeability, then we would expect the *pilE* mutant to have increased sensitivity to other antibiotics, which we did not observe (Table 2).

There are several possibilities for how piliation could affect iron homeostasis (**Fig 5**). The pilus could directly sequester iron extracellularly and limit intracellular labile iron by either binding to iron or transporting iron either in its free form or complexed to another compound (**Fig 5A** and **5B**). The pilus facilitates the entry of diverse substrates, including DNA, certain antibiotics, heme, and triton X-100 and, therefore, could transport iron [53]. Piliation may impact iron homeostasis through transcriptional changes (**Fig 5C**) of genes involved in iron transport or iron storage (**Fig 5D** and **5E**). In a microarray analysis of a mutant that lacked the retraction ATPase PilT, the transcription of 63 genes were altered, including an iron transporter gene *fetA* [54]. The effect of the pilus on iron is not found in every type IV pili expressing species as *Pseudomonas aeruginosa* does not exhibit pilus-dependent resistance to hydrogen peroxide [32]; therefore, pilin-dependent iron homeostasis depends on additional factors in *N. gonorrhoeae* that is absent in *P. aeruginosa* or there are factors in *P. aeruginosa* that interfere with pilus-mediated iron regulation.

Likewise, there are multiple ways for how piliation could affect sensitivity to hydrogen peroxide and LL-37. If piliation directly alters the level of free iron, the basal level of intracellular

**Table 2. Strains and plasmids.**

| Strain | Description | Reference/source |
| --- | --- | --- |
| N-1-60 | FA1090 multisite G4 mutant 1-81-S2 *pilE* variant, pilC2PLon | [66] |
| N-1-69 | An unmarked Δ*pilE* mutant (deletion from the 6th amino acid to the stop codon in *pilE* from Dr. Alison Criss) in N-1-60 | [66] |
| N-1-5 | FA1090 Tn#9/G4 mutant Avd-1 1-81-S2 pilE variant KanR | [68] |
| N-1-56 | IPTG-regulatable *pilE* variant RM11.2 in N-1-5 ErmR | [44] |
| N-4-47 | F62 | Lab stock |
| N-4-46 | FA19 | Lab stock |
| N-4-53 | *pilE* deletion from N-1-69 in N-4-47 | this study |
| N-4-54 | *pilE* deletion from N-1-69 in N-4-46 | this study |

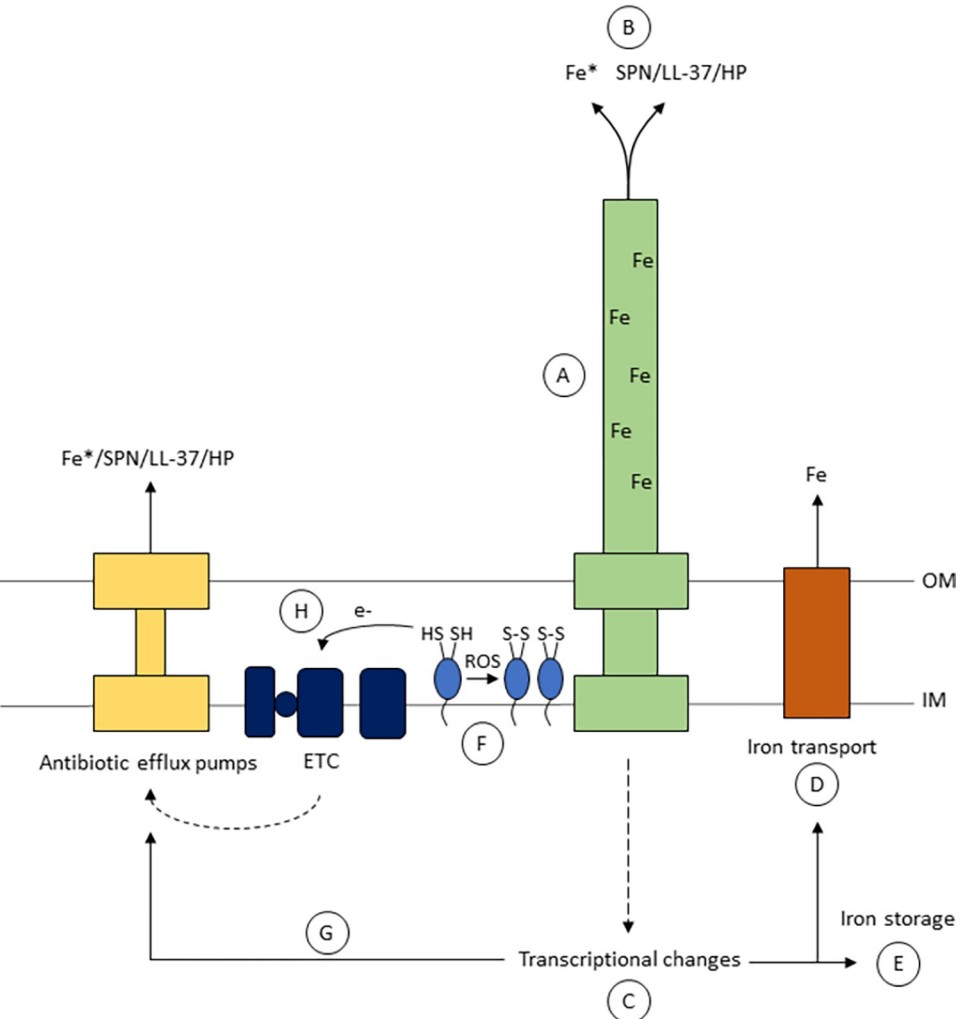

**Fig 5. Possible mechanism underlying pilin-dependent iron homeostasis and resistance to hydrogen peroxide and LL-37.** A. The pilus may bind to iron and/or B. transport hydrogen peroxide, LL-37, streptonigrin, and iron (Fe*) which could be in an unbound state or complexed with another substrate. C. Piliation could affect the transcription of genes that are important for D. iron transport, E. storage, or G. antibiotic efflux. F. Reduced pilin can quench reactive oxygen species and form disulfide bonds. H. Electrons from reduced pilin subunits can affect the electron transport chain to influence the activity of antibiotic efflux pumps. These pumps may transport streptonigrin, LL-37, hydrogen peroxide, and/or iron in its free or complexed form. Abbreviations for electron transport chain (ETC), inner membrane (IM), and outer membrane (OM) are used.

ROS would be affected through the Fenton reaction. The effect of piliation on labile iron can also impact antibiotic efflux. The ferric uptake regulator Fur affects the level of the repressor *mpeR* [55], which impacts the transcription of *mtrR* and *farR*, the transcriptional repressors of the *mtrCDE* and *farAB* operons, respectively [56–58]. Alternatively, reduced, free disulfide bonds on pilin subunits can also directly scavenge ROS, forming disulfide bonds (**Fig 5F**). Additionally, piliation may influence antibiotic efflux through the Mtr or Far systems (**Fig 5G**). It was reported that a *pilT* mutant increased expression of *mtrF* that encodes an inner membrane multiple transferable resistant protein [59]. This PilT-dependent effect occurs through the transcriptional repressor FarR that is also known to regulate the expression of the FarAB transport system [59]. Lastly, the effect of piliation on antibiotic efflux may be less direct. It was reported that an increase in NADH activated both the electron transport chain

activity and antibiotic efflux pumps in *P. aeruginosa* [60]. A pilin subunit in its reduced state with two free disulfide bonds could affect electron transport like NADH by acting as an electron donor and ultimately activating antibiotic efflux pumps in *N. gonorrhoeae* (**Fig 5H**).

There was no major difference in iron levels between piliated and non-piliated cells by ICP-MS. Most of the cellular iron is bound in iron-sulfur clusters. Given that streptonigrin and desferal can only access the labile iron pool, the pilus predominantly influences the level of free or loosely bound iron. If piliation only affects the labile iron pool, then the difference may be too small to measure the difference in the labile iron pool relative to the total iron in the cell.

We observed that the addition of iron or magnesium promoted survival in streptonigrin-treated *pilE* cells. We predict that the iron-dependent increase in streptonigrin resistance may be due to the repair of iron-sulfur clusters. The addition of magnesium may stabilize the outer leaflet of the outer membrane [61].

Since neutrophil oxidative antimicrobial activity is not required to kill *N. gonorrhoeae*, the antioxidative defense mechanisms may be particularly important during certain phases of infection. It has been proposed that these defense mechanisms may play a role early in the course of an infection [15,19] when gonococci encounter hydrogen peroxide from the epithelium [62] and commensal lactobacilli [63]. Alternatively, as an infection progresses, the immunostimulatory dead bacterial debris may outweigh the oxidative suppression of live cells, resulting in a strong PMN respiratory response. The pilus may, therefore, act as an antioxidant resistance mechanism that may also be sufficient to allow bacteria to survive and multiply at this stage of the infection [64,65]. Our results may explain why only piliated isolates are recovered from typically symptomatic clinical infections due to the protective effect of the type IV pili against PMN killing by limiting intracellular iron, promoting transmission, and continuing the chain of gonorrhea infections.

## Methods

### Strains and reagents

Strains used in this study are listed in **Table 2**. The parental strain N-1-60 [66] is *N. gonorrhoeae* strain FA1090 pilin variant 1-81-S2 (confirmed using primers pilRBS and SP3A (Table 3) [67]). N-1-60 carries multiple point mutations in the *pilE* guanine quadruplex sequence upstream of *pilE* [68] (confirmed with primers USS2 and pilAREV [66] (**Table 3**)) and the phase-locked "on" *pilC2* (NGO0055) sequence (confirmed using primers pilCfor and pilCdownstream (**Table 3**)), not *pilC1* as previously indicated [66]. For all strains, the 1-81-S2 *pilE* sequence was confirmed in PCR products using primers pilRBS and SP3A (**Table 3**) in addition to the non-variable guanine quadruplex and the phase-locked *pilC2*. This work was supported by the Northwestern University Sanger Sequencing Facility. Solid growth media was prepared with GC medium base (Difco) plus 1.25 g agar/liter, Kellogg supplement I [22.2 mM glucose, 0.68 mM glutamine, 0.45 mM cocarboxylase], and Kellogg supplement II

**Table 3. Oligonucleotides.**

| Name | Sequence (5'-3') | Source/reference |
|---|---|---|
| PILRBS | GGCTTTCCCCTTTCAATTAGGAG | [67] |
| SP3A | CCGGAACGGACGACCCCG | [67] |
| USS2 | TGAACCAACTGCCACCTAAGG | [68] |
| pilAREV | GGGCGGCAGTGTCGAAAATTGTCAGTTTTAGTGC | [66] |
| pilCfor | GGCGGAGGTGGCGGGGCC | [69] |
| pilCdownstream | CCATCTTTGGCGGTACCCTCGCTG | [69] |

[1.23 μM Fe(NO3)3] (GCB) at 37˚C in 5% $CO_2$. For liquid growth, cells were grown in liquid GCB media (GCBL) [containing 15 g/liter Bacto protease peptone 3, 23 mM potassium phosphate dibasic, 7.35 mM potassium phosphate monobasic, and 17.11 mM sodium chloride], Kellogg supplement I, and 5 mM sodium bicarbonate.

### Determination of FA1090 transformation efficiency

The efficiency of FA1090 carrying an IPTG-regulatable *pilE* transformation was performed similarly to the protocol that was described in reference [70] except that 50 ng of pSY6 plasmid DNA was used. After 20 minutes of incubation of the cells and DNA in the presence and absence of IPTG at 37˚C, 1 U DNase I was added to the transformation reaction mixtures, and they were incubated for 10 min at 37˚C followed by 4 hours of incubation with and without IPTG in 2 ml of GCBL in a 12-well plate. Cells were diluted and grown on GCB agar plates without antibiotic selection and plates that contained 1 μg/ml nalidixic acid. Transformation efficiencies are reported as the means from four independent experiments.

### Hydrogen peroxide sensitivity assay

Cells were grown on GCB solid medium from -80 C freezer stocks for approximately 18 hours at 37 C with 5% CO2. GCBL with supplement I and 5 mM sodium bicarbonate was inoculated to an $OD_{550}$ between 0.03–0.05 and grew for 2–3 hours, shaking at 220 rpm at 37˚C. Cultures were diluted to an $OD_{550}$ ~0.1 and grew for approximately 2 hours, shaking at 220 rpm at 37˚C or until the cultures reached mid-exponential phase $OD_{550}$ ~0.4–0.6. Cultures were normalized to an $OD_{550}$ = 0.07 and treated with a gradient of hydrogen peroxide (Sigma 323381) for 15 minutes at 37˚C with aeration either on a drum rotor or shaking at 220 rpm. For experiments using desferal, cultures were normalized to an $OD_{550}$ = 0.055 and pretreated with the desferal for 15 minutes at 37˚C and then exposed to hydrogen peroxide for 15 minutes at 37˚C in a 12-well plate. Relative survival was determined by serially diluting the cells onto GCB agar plates and comparing the number of CFU after hydrogen peroxide treatment to no hydrogen peroxide.

### LL-37 sensitivity assay

LL-37 (Peptide Sciences) stock solutions of 2.5 mg/ml or 556.42 μM was prepared in 0.01% (v/v) glacial acetic acid [26], stored at -20˚C, and serially diluted in water before use. Strains were grown similar to that described for growing cells for hydrogen peroxide sensitivity assays. Cells were then diluted to an $OD_{500}$ of 0.05 in GCBL (supplemented with Supp I and 50 mM sodium bicarbonate). Cells were treated with LL-37 for 30 minutes at 37˚C shaking at 220 rpm. In experiments that involve treatment with desferal, DMTU, or tiron, cells were exposed to these compounds for 15 minutes at 37˚C shaking at 220 rpm prior to LL-37 treatment. For Fig 1C and 1D, the cells were washed by centrifuging at 3381 rcf for 2 minutes at room temperature, resuspended with GCBL, and 10-fold serial dilutions were spotted onto GCB agar plates. Relative survival was calculated by comparing the number of LL-37 resistant CFUs to the total number of CFUs in the absence of LL-37. The results are the averages and standard error of the means and representative of independent experiments.

### Streptonigrin sensitivity assay

Cells were grown overnight on GCB plates at 37˚C and 5% CO2 were inoculated into GCBL (supplemented with Supp I and 50 mM sodium bicarbonate) at an $OD_{550}$ of 0.04–0.15 and grown shaking at 37˚C for 2–3 hours. To test if *pilE* expression was necessary for streptonigrin

resistance, either no IPTG or 1 mM IPTG was added to the media to induce *pilE* expression from the native locus during the three-hour incubation period. To determine if various metals affected desferal-mediated rescue, 1 mM desferal was mixed with increasing concentrations of metal salts for 5–10 minutes before being added this mixture to a culture diluted to $OD_{550}$ ~0.05 for 5 minutes. Cultures were treated with dimethyl sulfoxide (Sigma D2650) or streptonigrin from *Streptomyces flocculus* (Sigma S1014) for 30 minutes (or 1 hour for Fig 2A) at 37˚C with aeration. Except for in Fig 2A, the cells were washed by centrifuging at 3381 rcf for 2 minutes at room temperature, and resuspended with GCBL before 10-fold serial dilutions were spotted onto GCB agar plates. Relative survival was calculated by comparing the number of CFU that survived streptonigrin treatment to DMSO vehicle control. DMSO was kept at or under 0.5% v/v final concentration.

## Antibiotic sensitivity tests

Cells that grew overnight on GCB agar plates were swabbed into 1 ml of GCBL and normalized to an $OD_{550}$ of 0.05 and 100 μl of the cell suspension was spread and absorbed into GCB agar plates. An Etest strip (bioMérieux, Durham, NC) was laid on top and patted down to ensure contact between the strip and the surface of the growth medium. After incubating for 17 hours at 37˚C in the presence of 5% $CO_2$, zones of clearance were recorded. The results are the ranges from three biological replicates.

## Metal analysis

The parent and *pilE* mutant were grown to mid-exponential phase in 40–50 ml. Cells were pelleted and washed with phosphate-buffered saline at least two times. The wash buffer contained 10 mM of a membrane-impermeable iron chelator diethylenetriaminepentaacetic acid (also known as DETAPAC or DTPA) to remove extracellular iron. The pellet was dried in a heat block for one hour at 100˚C. The condensation within the tube was wicked away with a clean Kimwipe. The samples were acid digested by adding 250 μl of trace grade nitric acid (OmniTrace Ultra nitric acid NX0408) and 50 μl of trace grade hydrogen peroxide (Sigma 95321). The samples were incubated in a heat block at 65˚C overnight and then diluted with 4.7 ml of water to a final volume of 5 ml and analyzed by the Quantitative Bio-element Imaging Center at Northwestern University on a computer-controlled (QTEGRA software) Thermo iCapQ ICP-MS (Thermo Fisher Scientific, Waltham, MA, USA) operating in KED mode and equipped with an ESI SC-2DX PrepFAST autosampler (Omaha, NE, USA). Internal standard was added inline using the prepFAST system and consisted of 1 ng/mL of a mixed element solution containing Bi, In, 6Li, Sc, Tb, Y (IV-ICPMS-71D from Inorganic Ventures). Each sample was acquired using 1 survey run (10 sweeps) and 3 main (peak jumping) runs (40 sweeps). Instrument performance is optimized daily through autotuning followed by verification via a performance report (passing manufacturer specifications). The results were normalized to total protein in each sample was determined by the Pierce BCA Protein Assay Kit (Thermo Scientific 23227).

## Supporting information

**S1 Fig. Desferal does not affect hydrogen peroxide or LL-37 killing in the piliated parent strain.** Relative survival of the FA1090 parent strain N-1-60 to hydrogen peroxide or LL-37 after desferal treatment was determined. The lines indicate the paired samples. The average relative survival of the parent to hydrogen peroxide was 0.0059 with 10 mM desferal and 0.0059 without desferal. The average relative survival of the parent to LL-37 was 0.033 with 10 mM desferal and 0.045 without desferal.
(TIF)

**S2 Fig. The effect of various metals on desferal-mediated rescue of *pilE* mutant from strep-tonigrin killing.** The *pilE* mutant was treated with either desferal alone or desferal mixed with various concentrations of FeCl3, n = 3 (A), MgCl2, n = 3 (B), ZnCl2 n = 3 (C), or MnCl2 n = 3 (D) before streptonigrin killing. Relative survival was determined by calculating the ratio of streptonigrin resistant colonies to the total number of cells in the reaction. The averages and the standard error of the mean are shown.
(TIF)

**S3 Fig. Transformation efficiency of IPTG-regulatable *pilE* strain.** The transformation efficiency of the FA1090 strain carrying an IPTG inducible *pilE* was determined using pSY6, a plasmid that carries a *gyrA* point mutation. The number of nalidixic acid-resistant colonies was compared to the total number of colonies in four biological replicates. The average and standard error are shown.
(TIF)

**S4 Fig. The antioxidants DMTU and tiron do not affect sensitivity to LL-37 in a *pilE* mutant.** The effect of DMTU (A, n = 3) and tiron (B, n = 4) on LL-37 sensitivity in the *pilE* mutant. The mean and standard error of the mean are plotted and analyzed by Wilcoxon matched-pairs signed-rank test but are not statistically significant.
(TIF)

## Acknowledgments

We thank Wendy Geslewitz for the insightful discussions on this research and Brian Sands for his technical assistance. Metal analysis was performed at the Northwestern University Quantitative Bio-element Imaging Center generously supported by NASA Ames Research Center Grant NNA04CC36G.

## Author Contributions

**Conceptualization:** Linda I. Hu, Elizabeth A. Stohl, H Steven Seifert.

**Data curation:** Linda I. Hu, Elizabeth A. Stohl.

**Formal analysis:** Linda I. Hu, Elizabeth A. Stohl.

**Funding acquisition:** H Steven Seifert.

**Investigation:** Linda I. Hu, Elizabeth A. Stohl.

**Methodology:** Linda I. Hu, Elizabeth A. Stohl.

**Project administration:** H Steven Seifert.

**Supervision:** H Steven Seifert.

**Validation:** Linda I. Hu, Elizabeth A. Stohl.

**Writing – original draft:** Linda I. Hu.

**Writing – review & editing:** Linda I. Hu, H Steven Seifert.

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
