## [Decision Letter · Decision Letter 0]

8 Dec 2021

Dear Professor Seifert,

Thank you very much for submitting your manuscript "The Neisseria gonorrhoeae type IV pilus promotes resistance to hydrogen peroxide- and LL-37-mediated killing by modulating the availability of intracellular, labile iron" for consideration at PLOS Pathogens. As with all papers reviewed by the journal, your manuscript was reviewed by members of the editorial board and by several independent reviewers. In light of the reviews (below this email), we would like to invite the resubmission of a significantly-revised version that takes into account the reviewers' comments.

The reviewers acknowledged the quality of this work. But they also pointed out that: (i) no clear mechanism is identified as to how pili alter iron homeostasis and (ii) the results could be discussed further. I also encourage the authors to pay particular attention to points 14 and 15 of reviewer #1.

These points should be addressed before resubmission.

I apologize for the duration of the review process, which was necessary to produce an effective evaluation of this work.

We cannot make any decision about publication until we have seen the revised manuscript and your response to the reviewers' comments. Your revised manuscript is also likely to be sent to reviewers for further evaluation.

Sincerely,

Mathieu Coureuil

Guest Editor

PLOS Pathogens

Xavier Nassif

Section Editor

PLOS Pathogens

Kasturi Haldar

Editor-in-Chief

PLOS Pathogens

orcid.org/0000-0001-5065-158X

Michael Malim

Editor-in-Chief

PLOS Pathogens

orcid.org/0000-0002-7699-2064

The reviewers acknowledged the quality of this work. But they also pointed out that: (i) no clear mechanism is identified as to how pili alter iron homeostasis and (ii) the results could be discussed further. I also encourage the authors to pay particular attention to points 14 and 15 of reviewer #1.

These points should be addressed before resubmission.

I apologize for the duration of the review process, which was necessary to produce an effective evaluation of this work.

Reviewer's Responses to Questions

**Part I - Summary**

Reviewer #1: The manuscript by Hu et al addresses the mechanism as to why piliated cells of N. gonorrhoeae are more resistant to oxidative killing by hydrogen peroxide and to killing by the antimicrobial peptide LL-37. The manuscript builds on previous results of the authors showing that: i) the metalloprotease MPG (NGO1686) is highly expressed in response to sublethal levels of hydrogen peroxide, ii) MPG is required for full piliation of gonococcal strains and the latter is required for resistance to hydrogen peroxide and LL-37 killing. N. gonorrhoeae is provided with multiple genetic mechanisms to resist oxidative damage. This mechanistic redundancy highlights the importance of resistance to oxidative killing for pathogenesis. However, very often the mechanism by which these gene products afford resistance to oxidative damage is unknown. Base on this, the theme of the paper is highly relevant in the context of gonococcal pathogenesis. However, there are some problems, which can be addressed editorially, in the way data is presented and the conclusions drawn from the results (see below). This reviewer believes that although evidence is presented to suggest that the way full piliation protects cells from hydrogen peroxide oxidative damage is by restricting the intracellular pool of labile iron, an additional experiment is required to fully validate the evidence as conclusive (see point 14 below).

Reviewer #2: Hu Linda I. et al is a manuscript supporting evidence that the piliation state affects iron homeostasis in N. gonorrhoeae and influences the sensitivity to various antimicrobial agents, including hydrogen peroxide and the antimicrobial peptide LL-37. The topic is of interest to the field and is well done; The manuscript is excellent written, methods are robust and contain appropriate controls.

Reviewer #3: This report investigates the interesting observation that gonococcal pili protect against ROS and CAMPs by showing that there is a pilus-dependent effect on iron homeostasis. This is an interesting aspect of bacterial pathogenesis that is not well-explored and the data broaden the importance of the multi-functional type 4 pili in N. gonorrhoeae (Ng)

Strengths of the manuscript include the straightforward description of experiments and the way the data are clearly shown in the figures. The authors also test numerous mutants to explore their hypothesis. The results show that the process is likely multi- factorial, which makes defining the mechanism more difficult. This manuscript could be improved in terms of impact, however, if the authors at least provided a model that shows all the factors that might be involved and discussed each possible factor in more detail.

**Part II – Major Issues: Key Experiments Required for Acceptance**

Reviewer #1: Major editorial changes required:

1. In figure 1C and D the pilE mutant should be presented side-by-side with the wild type, so that one can compare the magnitude by which desferal rescue survival in the mutant and the wild type. Given that hydrogen peroxide requires iron in order to kill cells, addition of an iron chelator (desferal) is expected to protect any type of cell; therefore, the problem here is that without comparing the magnitude of protection between mutant and wild type one cannot conclude whether sensitivity of the mutant is due to iron content.

2. Similar to point 1 above, in figure 2B the side-by-side comparison with the wild type is not presented. In fact, in line 169 the authors discuss the result of desferal and streptonigrin treatment on the wild type strain, but I could not find the result..

3. In figure 2C the addition of extra iron seems to increase survival to streptonigrin in the absence of desferal (results discussed in text in line 179-181), while extra iron in the presence of desferal lowered survival (expected). This result seemingly contradicts the logic of figure 2B in which desferal protects mutant cells from streptonigrin killing. The author should explain why extra iron in the absence of desferal seems to increase survival. Perhaps statistical comparisons between the ±iron without desferal conditions and ±iron with desferal conditions are required to understand the significance of this result.

4. The result describing the survival in the presence of other metals as presented in line 182 is not represented anywhere. PloS Pathogens does not allow “Data not shown”; hence, the authors should show the results in a supplemental figure.

5. The authors do not explain why the survival ratio (approx. 6E-4 in Fig.3) of the IPTG-inducible pilE mutant without IPTG seems to be significantly lower than the pilE mutant strain (N-1-69) in figure 2B. Perhaps the authors should present results from both strains in the same graph, or comment about this difference.

6. I may have missed it, but in line 218 the authors present figure 3B which is not in the manuscript.

7. In figure 4 the authors did not find any statistical differences among different mutations with the isogenic wild type in relation to streptonigrin resistance (lines 218-220); however, they concluded that there is a biological effect on the mutations. Thus, the importance of this finding is highly questionable.

8. In line 251, the authors state that the WT result is represented in figure 6, but it is not. As in point 1 above, the WT should be represented side- by-side with the pilE mutant to compare the magnitude of protection granted by the antioxidants against streptonigrin.

9. The result for the total iron content described in line 269 is not represented anywhere.

10. In line 286-287 author discuss the effect of other ROS scavengers, but the result is not presented. Again, the data should be shown in a supplemental figure.

11. In lines 297-299 the explanation about the higher iron influx, which can fuel the Fenton reaction (and the consequent production of hydroxyl radicals and oxidative damage) in the pilE mutant seems to contradict the discussed result of figure S2 in which LL37 killing is not affected by addition of antioxidants.

12. In line 332 the authors suggest that “desferal could only access labile iron pool”. Do the authors know for sure that desferal can penetrate gonococci? If the authors have a reference in which it is shown that desferal makes it through the cell membrane, they should add it to the text.

13. The authors should consider a summary figure that highlights the proposed mechanism of HP and LL-37 resistance with respect to pili and labile iron.

Major technical concerns:

14. The authors did not find any difference in total iron content by ICP-MS. Perhaps the difference between wild type and pilE mutant cells is where the iron is stored. If pilus bundles have some iron-chelation properties, then the wild type strain would accumulate iron at the cell surface or perhaps in the periplasm while the mutant would transport it directly into the cells. Have the authors considered performing the ICP-MS assay from gonococcal spheroplasts? Another alternative is to measure directly labile iron content using the colorimetric unified-ferene (u-ferene) assay, reported elsewhere. These experiments would help in supporting the conclusion that the way full piliation protect cells from hydrogen peroxide killing is by restricting the intracellular pool of labile iron.

15. Pyruvate can be a scavenger of toxic radicals produced by the Fenton reaction. Thus, the authors should consider testing the susceptibility of gonococci to HP and LL-37 when grown in pyruvate or by including pyruvate in the killing assays.

Reviewer #2: Additional experiments required:

- Figure 5: the authors should show data from three independent experiments for FA19.

Reviewer #3: Only one experiment is suggested (see below); the main issue is a need to at least help the reader visualize how the many factors hypothesized to play a role, based on the mutant testing, may do so, in absence of one or two easily tested mechanisms.

The authors should be commended for testing such a large number of mutants to explore their hypothesis. Results from the testing of pilus biogenesis genes and the complementation data with an inducible pilE gene support the hypothesis that the presence or assembled pili modulates intracellular iron. The conclusion that the effect of pili on streptonigrin killing is multifactorial also appears likely based on mutant data; however, more discussion of how these different is needed. For example, the results with the penicillin-binding protein mutants are interesting but more discussion would help the reader visualize what might be going on at the level of cell wall integrity or whatever the authors are thinking. No mention of the mpg gene is made after the introduction, and since Mpg also can affect PG, a more detailed discussion of the role of these three genes (dacB, dacC and mpG) in PG biosynthesis and/or degradation, and how their loss might lead to changes in iron homeostasis would increase the value of the data.

Perhaps one possible mechanism could be tested: The authors state “Based on all the data presented, we propose that the differential sensitivity to streptonigrin, hydrogen peroxide, and LL-37 is due to piliation reducing the intracellular labile iron pool. This would be a possible explanation if piliation modulates iron import and, in the absence of the pilus, iron influx is uninhibited, leading to differential killing in the pilE mutant”. The authors showed that nonpiliated Ng do not have increased membrane permeability. Can the hypothesis that piliation modulates iron import be tested?

The authors mention pilin-dependent iron homeostasis (line 311). Some discussion about what is known about iron homeostasis in Ng is needed. Is there a known homeostasis system? Are there transcriptional regulators that alter this process? Is it known how ferrous iron enters the cell?

Numerous possibilities are listed in the discussion on page 17 as to how pilin or pilus may protect against ROS and elsewhere, how pili may directly or indirectly influence intracellular labile iron stores. A figure showing a model of all the possible forces that might lead to increased intracellular iron in nonpiliated Ng [– all of which are listed within the text in the discussion section (~lines 302 – 329)] - would help the reader think more about different mechanisms that contribute to this process (i.e. transcriptional networks (i.e. published data that are mentioned in the text), cell wall changes, direct or indirect of pilin on the electron transport chain (can this be tested?), efflux differences)

**Part III – Minor Issues: Editorial and Data Presentation Modifications**

Reviewer #1: 1. Line 101 correct “predominantly”

2. Line 373 describe the concentration unit of glutamine

3. Line 373 double check concentration of Fe(NO3)3 in GC broth. In our GC broth recipe is 12.3 µM the final concentration.

4. Line 126 it should be panel D not C

5. The syntax of sentence in line 132 is hard to understand

6. Line 212 delete “for”

7. Improve sentence writing in line 213. It is not easy to read

8. Declare what the acronym of “ROS” stands for

9. Line 245 add “the” after “through”

10. Line 250 change “then” to “when”

11. Line 292 change “with” to “by”

12. Please clarify if N-1-60 is isogenic to FA1090.

Reviewer #2: General comment:

I would like to recommend to the authors to shorten the introduction and to focus better on the subject, which is dealt with in the manuscript. For example, the paragraph beginning at line 44 "N. gonorrhoeae primarily..." until line 54 could be deleted in its entirety.

Minor comments:

- Page 6, line 91 and throughout the whole manuscript (e.g. line 198): type IV pili should be used consistently throughout the document or abbreviated

- Figure 2: the authors should name the concentration of DMSO used or mention it in the material and methods section

- Figure 4: have the authors characterized the piliation status of the different mutants phenotypically (Immunoblotting, dot blot, EM)?

- Paragraph: Total iron content is unaffected by piliation : the authors mention that the parental strain and the pilE mutant showed no significant difference in total iron . data should be shown if possible (for example as Suppl figure S3)

Reviewer #3: Minor corrections

Page 4 “In men, infections can cause inflammation of the epididymis or epididymitis”. As written it sounds like epididymitis is a different condition than inflammation of the epididymis.

Page 4: “These local gonococcal infections can also enter the bloodstream to develop disseminated gonococcal infection” The word “local” should be changed to “localized” or “mucosal”. Ascending infections (upper reproductive tract, male or female) aren’t actually considered localized (they are “locally disseminated” infections that can involve 2-3 organs, and so “mucosal” is the best adjective.

Page 5 “Even in the presence of a bactericidal oxidative response when N. gonorrhoeae expressing outer membrane opacity (Opa) proteins variants can engage specific receptors on PMNs [25]… “This is a complicated sentence that may lose the reader who is not familiar with Ng Opa proteins/CEACAMS on neutrophils, etc.

Page 8 Vibrio cholera should be Vibrio cholerae

Page 34 “… determined using pSY6, a plasmid that carries a gyrA point mutant” - mutant should be mutation.

PLOS authors have the option to publish the peer review history of their article (what does this mean?). If published, this will include your full peer review and any attached files.

Reviewer #1: No

Reviewer #2: No

Reviewer #3: No
---

## [Decision Letter · Decision Letter 1]

29 Apr 2022

Dear Hank,

We are pleased to inform you that your manuscript 'The Neisseria gonorrhoeae type IV pilus promotes resistance to hydrogen peroxide- and LL-37-mediated killing by modulating the availability of intracellular, labile iron' has been provisionally accepted for publication in PLOS Pathogens.

Best regards,

Mathieu Coureuil

Guest Editor

PLOS Pathogens

Xavier Nassif

Section Editor

PLOS Pathogens

Kasturi Haldar

Editor-in-Chief

PLOS Pathogens

orcid.org/0000-0001-5065-158X

Michael Malim

Editor-in-Chief

PLOS Pathogens

orcid.org/0000-0002-7699-2064

Reviewer Comments (if any, and for reference):

Reviewer's Responses to Questions

**Part I - Summary**

Reviewer #1: The authors have responded to my previous concerns and modified their paper accordingly. Critically, they now propose a model, which will require additional experiments in the future io test. While I am satisfied with the revisions and find the paper to be of high quality, I would like to suggest a mechanism that involves transcriptional control of the mtrCDE efflux pump and availability of free iron. In this respect, evidence has been presented that Fur+Fe regulates the level of the MpeR repressor. MpeR represses expession of mtrR, which encodes a repressor of the mtrCDE efflux pump operon, and this could result in increased levels of the MtrCDE efflux pump thereby impacting levels of LL-37 susceptibility.

Reviewer #2: The manuscript is excellently written, the topic is of interest to the field, and the experiments are well done; the methods used are robust and include adequate controls.

All objections and criticisms noted have been addressed to my complete satisfaction.

Reviewer #3: This manuscript by Hu et al. is a follow-up on the previous observation reported by this group that piliation protects Neisseria gonorrhoeae (Ng) from H2O2, cathelicidins, and non-oxidative PMN killing. This work has several merits including further investigation of this finding, the results of which expand the impressive list of pilus-mediated functions in this pathogen.

Other strengths include the clear presentation of the data, the testing of metals in addition to iron, complementation of the pilE mutants, examination of intracellular iron pools, and the demonstration that the resistance to both ROS and CAMPs afforded by piliation is iron-dependent, but ROS are not involved in the latter. The authors conscientiously responded to the reviewers’ critiques including the addition of Figure 5 to provide a model for the reader to put this all together, which was a good suggestion by the reviewers and helps round out the manuscript.

**Part II – Major Issues: Key Experiments Required for Acceptance**

Reviewer #1: None

Reviewer #2: no further objections

Reviewer #3: None

**Part III – Minor Issues: Editorial and Data Presentation Modifications**

Reviewer #1: None

Reviewer #2: no further objections

Reviewer #3: One minor correction: Lines 47-49 – should read: In women, infection spreads from the cervix to other areas of the reproductive tract to cause pelvic inflammatory disease (the cervix is the primary site of infection, not the vagina; Ng is not a vaginal pathogen in women of reproductive age)

PLOS authors have the option to publish the peer review history of their article (what does this mean?). If published, this will include your full peer review and any attached files.

Reviewer #1: No

Reviewer #2: No

Reviewer #3: No

---

## [Editor Report · Acceptance letter]

13 Jun 2022

Dear Professor Seifert,

We are delighted to inform you that your manuscript, "The *Neisseria gonorrhoeae* type IV pilus promotes resistance to hydrogen peroxide- and LL-37-mediated killing by modulating the availability of intracellular, labile iron," has been formally accepted for publication in PLOS Pathogens.

Best regards,

Kasturi Haldar

Editor-in-Chief

PLOS Pathogens

orcid.org/0000-0001-5065-158X

Michael Malim

Editor-in-Chief

PLOS Pathogens

orcid.org/0000-0002-7699-2064